# $R^2$: Range Regularization for Model Compression and Quantization

## Abstract

Model parameter regularization is a widely used technique to improve generalization, but, it can also be used to shape the weight distributions for various purposes. In this work, we propose range regularization ($R^2$) for building quantization and compression friendly models by removing outliers from weights during training. By effectively regulating range of weights, we mold the overall distribution into a tight shape to ensure high quantization bit resolution, therefore allowing model compression and quantization techniques can to utilize their limited numeric representation powers better. We introduce $L_\infty$ regularization, its extension margin regularization and a new soft-min-max regularization to be used as a regularization loss during full-precision model training. We show that this technique generalizes well for post training quantization, quantization aware training methods like EWGS and compression techniques like DKM. Coupled with state-of-the-art quantization and compression techniques, models trained with $R^2$ perform better on an average, specifically at lower bit weights with 16x compression ratio. Our results show that $R^2$ generates state of the art 2-bit quantized models for heavily parameter constrained models like MobileNet V1 and V2 when coupled with EWGS. Additionally, for high compression ratio (32x), models trained with $R^2$ significantly better than the ones trained without it.

## 1 Introduction

Deep neural networks have become popular in human computer interaction, anomaly detection, financial markets, etc. Since a lot of applications running these models run on edge devices, running these compute-intensive models requires a balance of accuracy, latency, power efficiency and size for these models.

Quantization of neural networks is an effective way of reducing the power, latency, and size of these models. This requires that these quantized models are executed on specialized platforms with low-bit supports and at the cost of accuracy drop Dong et al.; Wang et al. (2019). Post training Quantization involves distributing all the available weights in a layer into bins spread equally apart across the range. Training time quantization techniques Bengio et al. (2013); Zhou et al. (2016), use stochastic gradient descent to quantize and optimize the weights (i.e., mapping each floating point number to a set of discrete bins according to the precision target). Quantization bit-resolution is inversely proportional to the range of weights and affects accuracy of the quantized models. Since outliers tend to increase range, outliers are detrimental for quantization friendly models.

As an example, lets assume we want to quantize the weight distributions shown in Figure 1 into 3 bins. For the distribution in black (original distribution) most of the weights will be quantized to zero and the model accuracy would drop significantly. This problem gets worse for low bit quantization and compression such as one or two bit quantization.

We introduce **Range Regularization** ($R^2$), a simple yet powerful method that helps to remove outliers during training without severely affecting full precision accuracy and provides a quantization or compression friendly checkpoint. Using range regularization we intend to trim the edges of the black distribution and convert it to a distribution similar to the one shown in red in Figure 1. We propose 3 different formulations of $R^2$ and through experimental analysis show that models trained

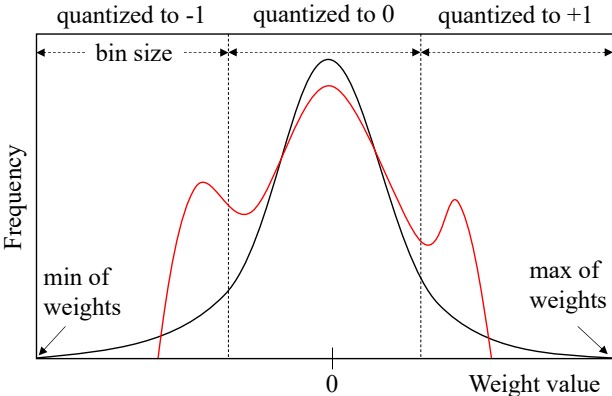

Figure 1: Example weight distribution for weight quantization with three bins. Black: without a range regularizer and Red: with a range regularizer.

with $R^2$ are more quantization and compression friendly. We also show that $R^2$ is invariant to the final quantization or compression algorithm.

Our experiments on MobileNet-V1 Howard et al. (2017) and MobileNet-V2 Sandler et al. (2018) using post training quantization techniques like DFQ Nagel et al. (2019) shows $> 10\%$ imrpovement using $R^2$ trained models. For QAT methods like EWGS Lee et al. (2021) $R^2$ trained models are roughly 4% better than the ones trained without $R^2$ for 2bit weight and activation quantization. For models compressed using 32x compression, $R^2$ improves the accuracy of parameter constrained models like MobileNet-V1 by 5% (2-bit,2-dim) and by 1.5% (4-bit,4-dim). We have also extended $R^2$ to fine-tuning MobileBERT on QNLI task, where we see an absolute 2% improvement in accuracies for 1-bit and 2-bit compressed models.

## 2 RELATED WORKS

### 2.1 MODEL COMPRESSION

The simplest and one of the most effective form of compression involves sharing weights within a layer. Deep Compression Han et al. (2016) introduced k-means clustering based weight sharing for compression. Initially, all weights belonging to the same cluster, share weight of the cluster centroid. During forward, loss is calculated using the shared weights which are then updated during backward pass. This leads to a loss of accuracy and model train-ability because the weight to cluster assignment is intractable during weight update Yin et al. (2019). DKM Cho et al. (2021) introduces differentiable k-means clustering, therefore making cluster assignments tractable. During forward clustered weights are used, however during backward the gradient is applied on the original weights.

### 2.2 MODEL QUANTIZATION

Model quantization reduces the memory footprint of a model by reducing the representative bits per weight for a given model. It also quantizes activation values so that we can convert floating point computation into integer computation which gives us a benefit of hardware efficiency. In this paper we have applied our $R^2$ with various training time quantization (quantization-aware training, QAT) algorithms like EWGS Lee et al. (2021), LSQ Esser et al. (2019) and DoReFa Zhou et al. (2016) used in PACT Choi et al. (2018). PACT clips activation values with a trainable parameter for activation quantization and uses DoReFa for weight quantization. LSQ quantizes weights and activations with learnable step size (scale or bin size). EWGS applies gradient scaling with respect to position difference in between original full precision weights and quantized weights based on LSQ.

Also, we have compared our $R^2$ with a tate-of-the-art post-training quantization (PTQ) methods. DFQ Nagel et al. (2019) equalizes per-channel weight ranges by applying per-channel scaling fac-

tors. It resolves the wide weight range problem across channels, but still the weight range would remain wide for lower bit quantization like 4bit as DFQ does not target outliers within a channel. In our experiment, models trained with our $R^2$ can be effectively quantized to 4bit weight / 8bit activation by PTQ without DFQ. AdaRound Nagel et al. (2020) proposed adaptive rounding for quantization bin assignment instead of nearest rounding. Pre-trained models with $R^2$ also show better quantization accuracies with AdaRound than models trained with just L2 norm which is well-known regularization.

In our extensible experiments, we show our $R^2$ improves accuracies with cutting-edge QAT and PTQ for lower bit quantization like 2bit weight / 2bit activation and 4bit weight / 8bit activation.

### 2.3    REGULARIZATION FOR QUANTIZATION

As we mentioned earlier, range regularization is not a quantization or compression technique. It helps compression and quantization by removing outliers in the weight distribution. Shkolnik et al. (2020) show that uniform distribution of weights are more robust to quantization than normally-distributed weights. To this they propose KURE (KUrtosis REgularization) to minimize the kurtosis of weights and ensure a uniform distribution. This method is independent of the quantization bit-width, therefore supports PTQ (Post-Training Quantization) in addition to QAT (Quantization Aware Training). However, this method is best suited for generic models which need to be specifically tuned for a given bit precision use case. To reduce the accuracy drop due to quantization Han et al. (2021) proposes to constrain weights to predefined bins based on the quantization bit-width. However, selecting these bins is a difficult process and the output of the models in very sensitive to the bin selection. In addition to that these methods ignore the effect of quantizing the first and last layers of the models.

## 3    RANGE REGULARIZATION

We introduce $R^2$ as a regularization technique to reduce the range of weights for every layer to get better pre-trained models for further quantization or compression. Just like $L_1$ and $L_2$ regularization our approach is invariant to the quantization or compression technique used. But as opposed to $L_1$ or $L_2$ regularization, $R^2$ only affects the outliers in weights by penalizing them while maintaining accuracy of the full precision model. We demonstrate that $L_2$ regularization (1x and 10x) does not solve the problem of outliers in Figure 2. As a reference for the expected weight distribution for a quantization friendly model we use the weight distribution from a model trained using KURE Shkolnik et al. (2020). While, the idea of minimizing range can be formulated in various ways we propose 3 different ways of defining $R^2$. We start from $L_\infty$ loss, extend it to margin loss and finally introduce soft-min-max loss for adding $R^2$ to the training loss.

$L_\infty$ **regularization**: This method tries to penalize only the outliers in an iterative manner during training by adding $L_\infty(W)$ as a regularization term for every layer in the model.

$$L_{reg} = \sum L_\infty(W) \tag{1}$$

The effect of this formulation is described in Figure 2 where it shows that $R^2$ helps to get rid of all outliers in the model. In addition, it brings the overall range of weight down in contrast to KURE Shkolnik et al. (2020), while, also making the weight distribution similar to a mixture of Gaussians as seen in KURE.

**Margin range-regularization**: This is an extension of $L_\infty(W)$ regularization, where, we define a margin for the range of allowed weights. Any weight outside this margin is penalized. In addition, we also penalize the width of the margin to ensure that the range of the overall weight distribution is small. The regularization loss for a given weight W is shown in Equation (2). Here $M$ is a learnable parameter per layer.

$$L_{reg} = \sum(|M| + \max(|W| - |M|, 0)) \tag{2}$$

The effect of margin regularization is similar to that of $L_\infty$ in terms of the final weight distribution as evident from Figure 2. The only difference is that margin $R^2$ penalizes all weight outside the margin per iteration in contrast to penalizing only the maximum weight.

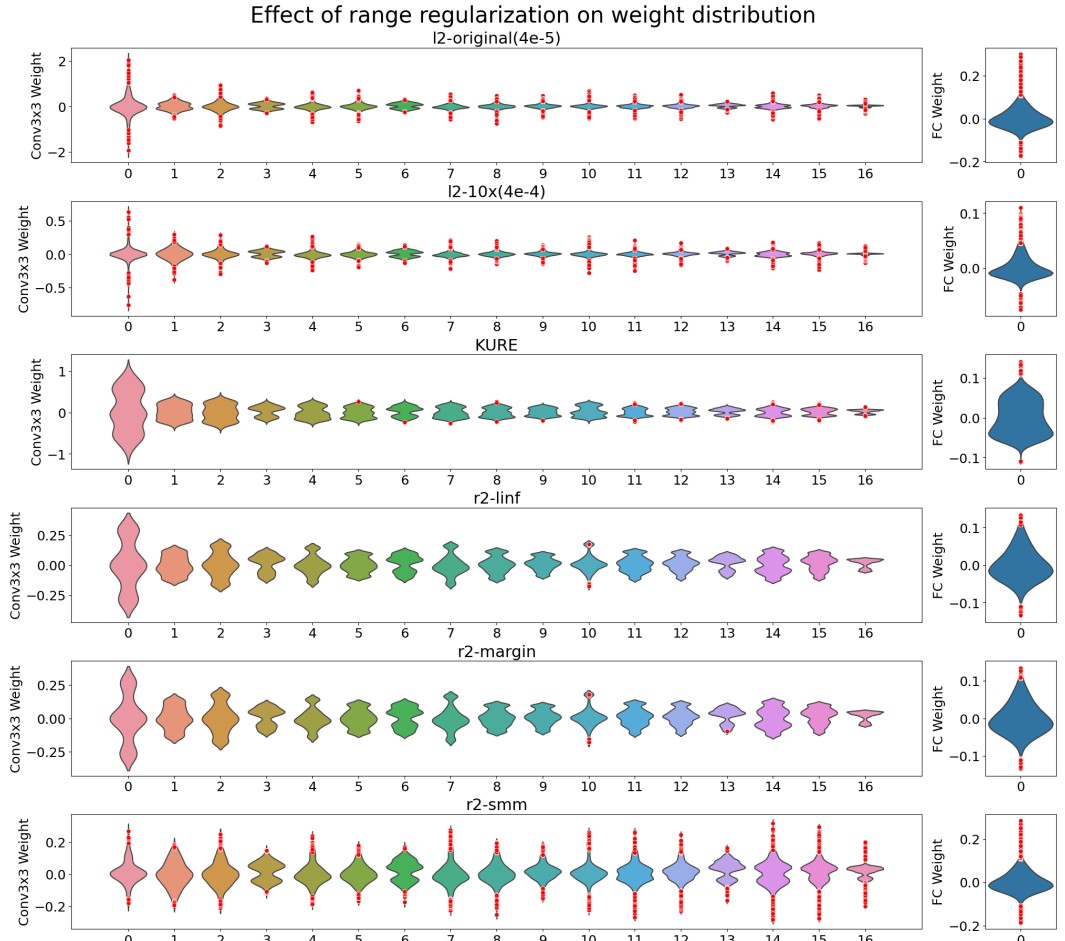

Figure 2: Weight distribution of 3x3 convolution layers and FC layer of MobileNet-V2 using only L2 norm and the proposed methods (the red dots correspond outliers).

**Soft-min-max range-regularization** : In this approach, we propose an asymmetric range regularization, to eliminate the constraint on the magnitude of weights and strictly enforce it on the range of weights. We hypothesize that such a technique will improve asymmetrically quantized models using techniques such as DKM Cho et al. (2021). The loss for a given weight W is described in Equation (3).

$$
\begin{aligned}
s_{max} &= \frac{\Sigma(W \odot e^{\alpha \times (W - W_{max})})}{\Sigma e^{\alpha \times (W - W_{max})}} \\
s_{min} &= \frac{\Sigma(W \odot e^{-\alpha \times (W - W_{min})})}{\Sigma e^{-\alpha \times (W - W_{min})}} \\
L_{reg} &= (s_{max} - s_{min}) + e^{-\alpha}
\end{aligned}
\tag{3}
$$

Here temperature $\alpha$ is a learnable parameter per layer. $e^{-\alpha}$ term in the regularization loss $L_{reg}$, encourages temperature $\alpha$ to increase during training time optimization process to approach hard-min-max regularization towards the end of train time optimization. This loss does not specifically penalize the outliers rather brings the overall range of weights down. Therefore, it might be susceptible to outliers as seen in Figure 2.

All the above mentioned range-regularization techniques were employed during training time of the base model itself and not during quantization or compression. This was done because the purpose of the $R^2$ is to provide effective initial weights for compression or quantization. This ensures extensibility of range-regularization to any quantization or compression technique.

Table 1: Model size of compressed MobileNet-V1 (in M bytes). F: first layer, L: last layer. EWGS 2bit quantization accuracies on ImageNet excluding the first and last layer quantization and all layers quantization.

| BIT | QUANT EXCL. F & L | QUANT ALL |
|---|---|---|
| 32BIT | 16.1 | 16.1 |
| 4BIT | 7.1 | 4.2 |
| 2BIT | 5.6 | 2.2 |
| 1BIT | 4.8 | 1.2 |
| EWGS 2BIT ACC. | 59.10% | 55.96% |

## 4  EXPERIMENT

### 4.1  EXPERIMENT SETTINGS

#### 4.1.1  PRE-TRAINING FROM SCRATCH WITH RANGE REGULARIZERS

We train ResNet-18 He et al. (2016), MobileNet-V1 Howard et al. (2017) and MobileNet-V2 Sandler et al. (2018) on ImageNet 1K Deng et al. (2009) with proposed Range Regularizers on a x86 Linux machine with eight GPUs to get pre-trained models before model compression and quantization-aware training. We set initial learning rates to 1.0, 0.4 and 0.4 for ResNet-18, MobileNet-V1 and MobileNet-V2 respectively. We use SGD with 0.9 of momentum with Nesterov. We apply 1e-4 of weight decay (L2 norm weight regularization) for ResNet-18 and 4e-5 for MobileNet-V1 and V2. For higher L2-regularization, l2-10x, in Figure 2, we use 4e-4 of weight decay for MobileNet-V2 to see whether heavy L2-regularization helps quantization or not. Range Regularizer weights for each regularizers are set to 0.01. For Margin range regularizer, the margin threshold is initialized with 2x the standard deviation of the initialized weights. In Soft-min-max regularizer training, the learnable parameter $\alpha$ is initially set to 0.1. For comparison, we use pre-trained models of Resnet-18 from Torchvision. As Torchvision did not provide a pre-trained model for MobileNet-V1, we trained MobileNet-V1 without Range Regularizer using the same settings above. It can be observed from Table 3 that $R^2$ doesnt significantly affect the performance of the full precision models as well therefore provides a strong initial checkpoint for the model to be quantized. In addition, for a large model like ResNet-18 $R^2$ also shows a regularization effect similar to weight decay, therefore, better validation accuracy than the one without $R^2$.

#### 4.1.2  MODEL COMPRESSION AND QUANTIZATION

Range Regularizer is not a model compression nor quantization method. It regularizes weights during training of the base model from scratch. To evaluate the effectiveness of the regularizers with model compression and quantization, we apply state-of-the-art compression/quantization techniques, DKM Cho et al. (2021), LSQ Esser et al. (2019), EWGS Lee et al. (2021), Bias correction Nagel et al. (2019), DFQ Nagel et al. (2019), and AdaRound Nagel et al. (2020) to the pre-trained model with Range Regularizer. Except EWGS, since other works do not provide official implementation, we had to implement those techniques ourselves and for Bias correction, DFQ, and AdaRound, we used AI Model Efficiency Toolkit [1]. We follow the same hyper-parameters used in the works, but **we apply compression and quantization for all layers (and activations in case of quantization) including the first and last layers.** It is important to compress/quantize all layers including first and last layers considering computation burden at the first layer with a large convolutional filter size such as 7x7 convolutions in the first layer of ResNet and the large number of weights in the last linear layer, e.g., 1.2M of weights in the last layer of MobileNet-V1 which has 4.2M of weights in total. We have demonstrated this burden in Table 1 for more clarity. Due to the outliers and wide weight ranges in the first and last layers, quantizing all layers have less accuracy than quantizing a model excluding the first and layer layers as shown in Table 1. We rep-

---

[1] https://quic.github.io/aimet-pages/

Table 2: Top-1 accuracies (%) of MobileNet-V1 and V2 on ImageNet-1K using PTQ methods with 4bit weight and 8bit activation quantization. None: quantizing without any advanced PTQ techniques, DFQ Nagel et al. (2019), AR: AdaRound Nagel et al. (2020)

| PRE-TRAIN (FP32) | MOBILENET-V1 0.25 | | | | MOBILENET-V1 1.0 | | | |
|---|---|---|---|---|---|---|---|---|
| REGULARIZATION | FP32 | NONE | DFQ | AR | FP32 | NONE | DFQ | AR |
| BASELINE (L2) | 55.43 | 0.41 | 13.03 | 45.88 | 74.12 | 2.67 | 54.06 | 70.42 |
| BASELINE (10X L2) | 52.40 | 11.23 | 18.85 | 44.83 | 72.67 | 13.41 | 57.68 | 69.23 |
| KURE | 52.83 | 3.54 | 21.87 | 48.32 | 72.50 | 53.69 | 59.21 | 60.51 |
| R_LINF | 53.48 | 13.30 | 29.10 | 50.76 | 73.65 | **61.73** | 53.00 | **72.28** |
| R_M | 53.30 | **27.72** | **31.02** | **51.17** | 73.54 | **61.66** | 65.06 | **72.29** |
| | MOBILENET-V2 0.25 | | | | MOBILENET-V2 1.0 | | | |
| BASELINE (L2) | 53.90 | 0.73 | 17.27 | 49.49 | 73.08 | 2.57 | 56.56 | 71.29 |
| BASELINE (10X L2) | 49.25 | 0.96 | 0.10 | 45.12 | 71.00 | 4.17 | 0.09 | 68.06 |
| KURE | 52.80 | 15.98 | **27.39** | 50.96 | 72.49 | 39.21 | 62.49 | 71.68 |
| R_LINF | 53.40 | 13.08 | 26.51 | **51.68** | 72.64 | **59.95** | 62.39 | 71.76 |
| R_M | 52.82 | **31.11** | 27.00 | 51.12 | 72.73 | **60.03** | 66.04 | **71.89** |

resent $L_\infty$ range-regularization as $R\_Linf$, Margin range-regularization as $R\_M$ and Soft-min-max range-regularization as $R\_Smm$ in the results.

## 4.2 MODEL QUANTIZATION

### 4.2.1 POST-TRAINING QUANTIZATION, PTQ WITH $R^2$

We compare models trained using $R^2$ and other weight regularization, L2, heavy L2, and KURE Shkolnik et al. (2020) and quantized using PTQ methods such as DFQ Nagel et al. (2019) and AdaRound Nagel et al. (2020). There are two major techniques in DFQ, bias correction compensating bias in activation and cross-layer equalization applying scale factor per channel to make all channels in a layer have similar weight range. AdaRound adaptively rounds weights to quantization bins instead of naive nearest rounding.

As shown in Table 2, models trained with $R^2$ are more quantization friendly than other regularizations. KURE makes the weight distribution uniform, therefore, can reduce outliers as a side-effect while keeping a wide weight range. Therefore, KURE is more effective than L2 norm, but $R^2$ shows the best accuracy as it reduces outliers as well as weight range. On the other hand, heavy L2 regularization (10x L2) makes weight ranges smaller, but it does not remove outliers, therefore prove to be effective here.

Even without advanced PTQ approaches such as DFQ and AdaRound, models trained with $R^2$ can be reasonably quantized without any further fine-tuning. In Table 2, None with $R^2$ shows significantly higher accuracy than other regularization. The models with $R^2$ have good weight distribution already from pre-training so that they can be quantized with fairly high quantization accuracies.

Cross-layer equalization in DFQ does not help PTQ accuracy for models trained with KURE and $R^2$ as comapred to quantizing without any advanced PTQ approaches, None column in Table 2. It might be because models trained with KURE and $R^2$ do not have outliers therefore, per channel weight equalization is not required. Cross-layer equalization might make PTQ unstable for those models trained with $R^2$ as it propagates estimation errors from cross-layer equalization itself and bias absorption.

### 4.2.2 QUANTIZATION-AWARE TRAINING, QAT WITH $R^2$

We apply state-of-the-art quantization techniques like PACT Choi et al. (2018) while training the models from scratch using $R^2$. For other quantization aware training methods like, EWGS Lee

Table 3: Top-1 accuracies (%) of MobileNet-V1 and V2 on ImageNet-1K with EWGS varying model size. Weights and activations are quantized with the same bit.

| | MOBILENET-V1 | | | | | |
|---|---|---|---|---|---|---|
| WIDTH FACTOR | 0.25 | | 0.5 | | 1.0 | |
| NUMBER OF OF WEIGHTS (M) | 0.47 | | 1.33 | | 4.23 | |
| PRE-TRAIN / QUANTIZATION | FP32 | 2BIT | FP32 | 2BIT | FP32 | 2BIT |
| w/o $R^2$ | 55.43 | 20.85 | 66.51 | 38.77 | 74.12 | 55.96 |
| KURE | 52.83 | 22.16 | 63.99 | 39.37 | 72.50 | 57.80 |
| R_LINF | 53.48 | **26.04** | 65.68 | 41.22 | 73.65 | **59.05** |
| R_M | 53.30 | 24.35 | 65.83 | **42.61** | 73.54 | 58.41 |
| | MOBILENET-V2 | | | | | |
| WIDTH FACTOR | 0.25 | | 0.5 | | 1.0 | |
| NUMBER OF OF WEIGHTS (M) | 1.52 | | 1.97 | | 3.51 | |
| w/o $R^2$ | 53.90 | 27.82 | 65.55 | 39.25 | 73.08 | 53.93 |
| KURE | 52.80 | 26.86 | 64.64 | 38.51 | 72.49 | 53.97 |
| R_LINF | 53.40 | **30.56** | 65.09 | 42.55 | 72.64 | 56.35 |
| R_M | 52.82 | 29.61 | 64.86 | **43.78** | 72.73 | **57.36** |

Table 4: Top-1 accuracies (%) of ResNet18 on ImageNet-1K with various QAT methods. Weights and activations are quantized with the same bit. We used the pre-trained model without $R^2$ from torchvision.

| QAT METHOD | | PACT | | LSQ | | EWGS | |
|---|---|---|---|---|---|---|---|
| REGULARIZATION | FP32 ACC. | 2BIT | 4BIT | 2BIT | 4BIT | 2BIT | 4BIT |
| w/o $R^2$ | 69.76 | 51.97 | 66.90 | 58.33 | **69.90** | 65.42 | **70.19** |
| R_LINF | 70.15 | 55.26 | **68.45** | 62.23 | 69.55 | **65.72** | 70.17 |
| R_M | 70.08 | **56.24** | 68.30 | 62.25 | 69.56 | 64.27 | 69.77 |
| R_SMM | 69.84 | 55.64 | 68.36 | **62.47** | 69.45 | 64.94 | 69.80 |

et al. (2021) and LSQ Esser et al. (2019) we initialize the model to pre-trained ResNet-18 (RN) He et al. (2016), MobileNet-V1 (MN1) Howard et al. (2017) and MobileNet-V2 (MN2) Sandler et al. (2018) with range-regularization.

Table 3 shows 2 bit weight and activation quantization result of MobileNet-V1 and V2 using EWGS Lee et al. (2021) with various regularization such as w/o $R^2$ (only with L2 norm), KURE Shkolnik et al. (2020), and our range-regularization. For both the models with varying model sizes, $R^2$ outperforms the models trained with only L2 norm or KURE. Without $R^2$, accuracy of MobileNet-V1 2bit quantization using EWGS declines from 59.10% to 55.96% when we quantize all layers including the first and last layers as shown previosuly in Table 1. This is because the first and last layers have wide weight ranges and many outliers as shown in Figure 2. Our approach effectively reduces the outliers in the first and last layer which enables the 2bit quantized model to achieve similar accuracy to the case with original EWGS results where the first and last layer of the model remain in FP32.

As shown in Table 4, range-regularization helps the quantization techniques in improving their accuracy, especially for extremely low bit quantization such as at 2 bit while it shows similar accuracies with 4 bit. For example, all $R^2$s improve 2 bit quantization accuracy with LSQ to over than 62% from 58%, but there is no noticeable difference in 4 bit LSQ accuracies with and without $R^2$. The reason why range-regularization would not help much for higher bit like 4 bit quantization is that QAT can effectively represent outliers using many bits.

Table 5: Top-1 accuracies(%) of compression using DKM and range-regularization with ResNet18 (RN), MobileNet-V1 (MN1) on ImageNet.

| METHOD | $R^2$ | RN | MN1 |
|---|---|---|---|
| DKM 1-BIT, 1-DIM | NONE | 58.97 | 45.54 |
| | R_LINF | 59.52 | 49.74 |
| | R_M | **59.70** | 47.21 |
| | R_SMM | 59.27 | **52.58** |
| DKM 2-BIT, 1-DIM | NONE | 67.64 | 65.95 |
| | R_LINF | 68.53 | 67.06 |
| | R_M | 68.33 | 67.50 |
| | R_SMM | **68.63** | **67.62** |
| DKM 4-BIT, 1-DIM | NONE | 70.22 | 69.29 |
| | R_LINF | 70.34 | 69.43 |
| | R_M | 70.33 | 68.52 |
| | R_SMM | **70.52** | **69.63** |

Table 6: Top-1 accuracies(%) of compression using multi-dimensional DKM and range-regularization with ResNet18 (RN), MobileNet_V1 (MN1) on ImageNet.

| METHOD | $R^2$ | RN | MN1 |
|---|---|---|---|
| DKM 2-BIT, 2-DIM | NONE | 63.52 | 48.16 |
| | R_SMM | **64.64** | **53.99** |
| DKM 4-BIT, 4-DIM | NONE | 64.89 | 58.55 |
| | R_SMM | **66.10** | **60.05** |

Interestingly, soft-min-max regularization does not seem to be as good as $L_\infty$ or margin for quantization. As discussed in Section 3, soft-min-max regularization allows us to have an asymmetric weight distribution so that it would be more effective for model compression instead of symmetric model quantization.

## 4.3 MODEL COMPRESSION

We evaluate the effectiveness of range-regularization for compression with the state-of-the-art compression technique, DKM Cho et al. (2021), for ResNet-18 and MobileNet-V1. The bit-dim ratio, $\frac{b}{d}$ is an important factor in the DKM algorithm which effectively defines the kind of compression a DKM palettized model would see. We ran these experiments for both scalar and vector palettization. For scalar palettization($dim = 1$) we ran 1 bit, 2 bit and 4 bit compression. These experiments would yield roughly 32x, 16x and 8x compressed models respectively. Table 5 shows that range-regularization significantly improves accuracy from original scalar palettized DKM 1 bit, 2 bit and 4 bit models. As we discussed, there is no significant difference for higher bit compression like 4 bit because many bit compression can also cover outliers even with out $R^2$.

We also expand the application of range-regularization to vector palettization($dim > 1$) DKM Cho et al. (2021) as demonstrated in Table 6. For these experiments, we kept the effective bit-dim ratio, $\frac{b}{d}$ equivalent to 1 so as to see variation across the most heavily compressed model which would be close to 32x compressed. Since a vector palettized model will require range constraining for all dimensions, we applied multi-dimensional range-regularization for all layers that would be palettized during compression. For vector palettized ResNet-18 there is an average absolute improvement of $> 1\%$ using models pre-trained with range-regularization, and for vector palettized MobileNet-V1 it the gain ranges from 5% to 3%.

Finally we also validated that $R^2$ scales to other domains as well by applying it in compressing MobileBERT Sun et al. (2020). For Question Answering (QNLI) Rajpurkar et al. (2016) using

MobileBERT, the proposed regularization improved the performance of the model by 2% absolute as demonstrated in Table 7. Note that we applied $R^2$ to a QNLI fine-tuning task based on a pre-trained MobileBERT Wolf et al. (2020). It might be necessary to apply $R^2$ to the entire training task of MobileBERT from scratch so that $R^2$ would have more chances to get effective weight distribution for model compression. Through these experiments across a variety of tasks(Image Classification, Question Answering etc.) we can also see that the application of the proposed range-regularizers is task-invariant and yields solid results across domains.

Table 7: Question-answering NLI (QNLI) accuracies of MobileBERT using single dimension DKM

| METHOD | PRE-TRAIN | 1-BIT | 2-BIT |
|---|---|---|---|
| DKM | 90.41 | 61.34 | 80.12 |
| DKM + R_LINF | 90.66 | **63.17** | **82.13** |
| DKM + R_M | 89.09 | 61.80 | 80.98 |
| DKM + R_SMM | 90.83 | 61.49 | 80.87 |

## 5  CONCLUSION

In this paper, we introduced range regularization as an effective technique to get rid of outliers in the weight distribution during training. This serves as a good initialization for state of the art post training quantization, quantization aware training and compression strategies, therefore can be coupled with any of them. This helps to augment the accuracy gained from such techniques and is invariant to the quantization or compression algorithm. With the three proposed formulations of range-regularization we wanted to demonstrate that there can multiple approaches of defining $R^2$ and the final metric depends on these formulations. We also demonstrated how the proposed method converts a normally distributed weight to a more desired uniform distribution for model quantization. Finally, we establish that range regularization doesn't affect the performance of the floating point models as well and in some over parameterized models it also has a regularization effect to address over-fitting. We also observe that for some quantization techniques like EWGS the incremental benefit of applying range regularization is marginal especially at higher bits.

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
