# OpenReview forum: "$R^2$: Range Regularization for Model Compression and Quantization"
_ICLR.cc/2024/Conference — Submitted to ICLR 2024_

### Official Review · Reviewer_yMkf · 2023-10-28

**Soundness:** 3 good
**Presentation:** 3 good
**Contribution:** 3 good
**Rating:** 6
**Confidence:** 4

**Summary:**

This paper proposes a range regularization method for training quantized model. The range regularization extend from $\ell_\infty$ regularization to more advanced formulation that can remove the outliers in weight distribution, leading to less quantization error.

**Strengths:**

This paper proposes a novel regularization-based method to tackle the outliers in the quantized model. The method is straightforward and effective.

Experiments are conducted under different settings to show the effectiveness of the proposed method

The paper is well written and easy to follow.

**Weaknesses:**

There is no clear analysis on under which scenario each of the proposed range regularization formulation will outperform the others. It would be good to have a switching mechanism to decide which formulation to use.

Ablation study on the impact of regularization strength for the proposed regularizer would provide more insight on the stability of the proposed method.

**Questions:**

See weakness

---

> ### Author Response · Authors · 2023-11-23
>
> Thanks for raising interesting discussion topics regarding our paper.
>
> As you asked, it is important to decide which $R^2$ can be effective for which cases. We discussed that soft-min-max $R^2$ would be more effective for model compression and asymmetric quantization as it makes asymmetric weight distribution. In our quantization experiment, soft-min-max still outperforms without $R^2$ cases, but other $R^2$s show better accuracy than soft-min-max $R^2$. On the other hand, in our compression study, soft-min-max $R^2$ achieves higher accuracy than other $R^2$s.
>
> Then, let’s compare L-inf $R^2$ and Margin $R^2$. L-inf is a naive and simple $R^2$ to limit the weight range. Margin $R^2$  is advanced $R^2$ from L-inf as it applies learnable margin threshold. In Table 2 of our paper and the following tables for the $R^2$ strength ablation study with AdaRound and SQuant, Margain $R^2$ shows better accuracy in most of cases. Interestingly, Margin $R^2$ shows stable quantization accuracy across varying $R^2$ strength while L-inf $R^2$ has somewhat fluctuated quantization accuracy regarding the $R^2$ strength. We think that the learnable margin parameter makes more stable full-precision pre-training while L-inf always regularizes the most outlier only. Therefore, Margin $R^2$ (as well as Soft-min-max $R^2$) shows more stable full-precision accuracy before quantization. We would recommend to use Margin $R^2$ for symmetric quantization if it is hard to find proper $R^2$ strength for L-inf.
>
> Thanks for asking the ablation study regarding the $R^2$ strength. To make our study more rigorous, we conducted the ablation study with AdaRound and SQuant as shown in the following table. In the paper, we basically set the $R^2$ strength to 0.01.
>
> * **MobileNet-V2**
> * **Full-precision accuracy (without $R^2$: 73.08)**
> | $R^2$ strength |  0.1  | 0.05 | 0.01 | 0.005 | 0.001 |
> |----------------|:-----:|:----:|:----:|:-----:|:-----:|
> |    L-inf       | 68.37 | 71.65| 72.64| 72.89 | 73.10 |
> |    Margin      | 72.71 | 72.60| 72.73| 72.59 | 72.78 |
> |    S-min-max   | 71.51 | 72.50| 72.81| 73.01 | 72.88 |
>
> * **AdaRound result**
> | $R^2$ strength |  0.1  | 0.05 | 0.01 | 0.005 | 0.001 |
> |----------------|:-----:|:----:|:----:|:-----:|:-----:|
> |    L-inf       | 67.75 | 70.87| 71.76| 72.02 | 71.83 |
> |    Margin      | 71.84 | 71.83| 71.89| 71.85 | 71.91 |
> |    S-min-max   | 70.08 | 71.23| 71.77| 71.72 | 71.51 |
>
> * **SQuant result**
> | $R^2$ strength |  0.1  | 0.05 | 0.01 | 0.005 | 0.001 |
> |----------------|:-----:|:----:|:----:|:-----:|:-----:|
> |    L-inf       | 61.45 | 66.58| 63.98| 65.99 | 65.85 |
> |    Margin      | 65.04 | 65.75| 67.03| 68.57 | 67.30 |
> |    S-min-max   | 59.08 | 67.88| 66.86| 66.43 | 62.94 |

---

### Official Review · Reviewer_htAB · 2023-10-29

**Soundness:** 2 fair
**Presentation:** 3 good
**Contribution:** 2 fair
**Rating:** 5
**Confidence:** 5

**Summary:**

This paper posits that outliers in the weight distribution are the primary reason for the decrease in model accuracy after quantization. To address this issue, an R2 regularization method is introduced to constrain the range of model weights, thereby eliminating outliers and improving the accuracy of the quantized model.

**Strengths:**

This paper has a reasonable structure, simple and easy to understand methods, and is easy to read.

**Weaknesses:**

1. Many prior works have argued the impact of outliers on the quantization results, but this paper only references two from 2020 and 2021, and only references KURE from 2020 as the baseline for comparison, which lacks persuasiveness.
2. It is hard to prove that applying R2 can lead to higher accuracy after quantization than not applying it. First, only four quantization methods are discussed in the experiments, and they can not cover all the quantizers. Secondly, the existing experiments in the paper show that the results  w/o R2 can also achieve higher model accuracy than that with R2 (Table 4).
3. The work is incomplete or lacking contribution, and it would be better to include the design of a quantization method specifically for this range regularization in the paper. Clearly, EWGS itself already considers outlier handling, so using R2 leads to a decrease in accuracy, indicating the necessity of designing a corresponding quantization method for R2.

**Questions:**

1. I am interested in how the experimental results would differ if we applied the clip function instead of R2 regularization in model training.
2. The author mentions that R2 can be applied to PTQ, but R2 itself is a regularization method used during model training, while PTQ aims to avoid retraining. Do these descriptions conflict with each other?

---

> ### Author Response · Authors · 2023-11-23
>
> We appreciate your thoughtful review for our work.
>
> There are a couple of prior works tackled outliers in model quantization. We mainly compared $R^2$ with DFQ, PACT, LSQ, and EWGS in our paper as we thought they are representative prior works for outlier handing. DFQ focused that each channel has different weight ranges so that per-layer quantization suffers from a channel which has a very wide weight range. DFQ resolves the issue by applying channel-wise scaling factor to make each channel have similar weight ranges. From our experiment with DFQ (in Table 2), it seems that enforcing to reduce weight range at pre-training by $R^2$ is much better than DFQ approach. PACT highlighted outliers in activation, not weight. PACT removes outliers in activation by clipping them and the clipping thresholds are gradually adjusted as they are learnable parameters during quantization. LSQ and EWGS take a weight clipping approach. They clips weights at its initialization stage before starting a fine-tuning phase for quantization to deal with weight outliers.
>
> Also there are more recent works that consider outliers in model quantization as you mentioned such as [1] and [2]. [1] proposed activation smoothing to deal with outliers in activation of LLM. It applies smoothing factors for the outlier activation and applies another scaling factors for corresponding weights to compensate the outlier smoothing factors. It mainly focuses on activation outliers, not weight outliers. [2] proposed a similar concept of [1], but it also proposed token-wise clipping to deal with outliers. We may want to conduct more studies with those works in the context of activation outliers and LLM quantization/compression as one of our future works.
>
> EWGS basically takes +- 3 sigma of weight distributions in a pre-trained model as a clip point for each layer. As you suggested, we conduct an ablation study varying clip thresholds from +- 3 sigma to +- 1 sigma for MobileNetV2 2bit quantization using EWGS. Please note that +- 3 sigma covers 99.7% and +- 1 sigma covers 68.3% of weights assuming normal distribution of weights. With $R^2$, EWGS can achieve **57.36%** top-1 accuracy on ImageNet while weight clipping does not show noticeable difference varying clipping thresholds.
>
> * **MobileNet-V2 with 2bit weight/activation EWGS**
> | Clip threshold | 3 sigma | 2 sigma | 1 sigma |
> |-----|:----:|:----:|:----|
> | Top-1 Acc.     |  53.93  |  54.01  |  53.87  |
>
>
> We also would like to discuss about the concern that $R^2$ does not seem to improve quantization accuracy, especially for Table 4. As we discussed in the paper, we think that higher-bit quantization-aware training can cover the outlier as a higher-bit quantized model has enough weight representation power. Therefore, in 4bit QAT result, we can see with and without $R^2$ quantization shows almost the same quantization accuracy while 2bit QAT result clearly shows effectiveness of $R^2$ in Table 4. Please note that EWGS result in Table 4 seems somewhat marginal, but in Table 3, we can see that $R^2$ meaningfully improves EWGS accuracy for MobileNet-V1 and V2. So, we think again that weight clipping would not sufficient for handling outliers in model quantization.
>
> For the concern that we only showed accuracy improvement by $R^2$ with a limited number of quantization methods. As you pointed, we conducted the comparison study with DFQ, AdaRound, PACT, LSQ, KURE, and EWGS. There are bunch of different quantization methods and it is very hard to show the effectiveness of $R^2$ with all of them. To do our best, we choose diverse quantization methods in our study such as PTQ (equalizing per-channel weight range (DFQ), adaptive quantization bin assignment (AdaRound)) and QAT (activation clipping (PACT), learnable step-size (LSQ), gradient scaling (EWGS), weight regularization for stable quantization (KURE)). Also, we added a comparison study with more recent quantization works as shown in our response for the first reviewer vWZn, PD-Quant and SQuant. In the all comparison study, we can see the clear  evidence that $R^2$ helps quantization methods better at low-bit.
>
> PTQ does not require fine-tuning for quantization unlike QAT. We apply $R^2$ only for full-precision pre-training before PTQ. We propose that if someone considers PTQ, QAT, and/or compression, we would recommend to apply $R^2$ during pre-training from scratch. It rarely affects to full-precision model accuracy while it gives a quantization/compression friendly pre-trained model. In Table 2, 3, 4, we can see that full-precision model accuracies before quantization are similar with and without $R^2$.
>
> [1] Xiao, Guangxuan, et al. "Smoothquant: Accurate and efficient post-training quantization for large language models." International Conference on Machine Learning. PMLR, 2023.
>
> [2] Wei, Xiuying, et al. "Outlier suppression: Pushing the limit of low-bit transformer language models." Advances in Neural Information Processing Systems 35 (2022): 17402-17414.

---

### Official Review · Reviewer_aeAj · 2023-10-31

**Soundness:** 1 poor
**Presentation:** 2 fair
**Contribution:** 2 fair
**Rating:** 3
**Confidence:** 4

**Summary:**

In this paper, the authors introduce “range normalization,” a new technique for normalizing model parameters to form weight distributions suitable for quantization.

This approach acts as a regularization loss during full-precision model training, and shows that pre-training with R^2 improves accuracy in subsequent Post-Training Quantization (PTQ) or Quantization-Aware Training (QAT) steps. This means the possibility of obtaining highly generalized models.

The authors experimentally verified the applicability of the proposed method to other quantization techniques, such as EWGS or DKM. In particular, this study shows that when the proposed method is applied simultaneously with EWGS, state-of-the-art (SOTA) results can be obtained through 2-bit quantization on MobileNet V1/V2.

**Strengths:**

* This paper introduces a novel model parameter regularization technique called range regularization to shape a weight distribution conducive to compression/quantization.
* This paper shows that training a model from scratch using $R^2$ can result in a model with better accuracy after applying subsequent post-training quantization (PTQ) or quantization-aware training (QAT). Experimental evidence shows the potential to obtain more highly generalized models with this approach.
* The proposed method demonstrates its potential applicability to other quantization techniques, such as EWGS or DKM.

**Weaknesses:**

* Based solely on the experiments in the paper, it is challenging to discern whether the proposed $R^2$ method is more effective in terms of generalization compared to other regularization methods.
* While the proposed method demonstrated effectiveness in experiments with CNN-based models, it appears challenging to compare its efficacy in other tasks or architectures, such as Language Models, based solely on the conducted experiments.
* The paper demonstrates the effectiveness of $R^2$ based on weight distribution, but it does not address activations. Since the experiments in the paper show a significant compression ratio for activations, with activations also sharing the same number of bits as weights, it seems essential to include an analysis or discussion regarding activations.

**Questions:**

* The statement, 'In addition, for a large model like ResNet-18, $R^2$ also shows a regularization effect similar to weight decay, therefore, better validation accuracy than the one without $R^2$,' is unclear in its meaning. If $R^2$ exhibits a performance boost due to a regularization effect similar to weight decay, shouldn't $L_2$ regularization be more effective from the outset? A detailed discussion or experimental data addressing this is needed.
* From the experiments in the paper alone, it's difficult to ascertain the effectiveness of the proposed $R^2$ method in terms of generalization. Are there any experimental data available to assess the Generalization Gap?
* The paper only presents results for small models like CNN or MobileBERT, primarily focusing on CNN results. Are there any results showcasing the application of the proposed method to larger models such as ResNet-101 or LLM (e.g., transformer architectures like GPT-J)? Is there experimental evidence showing the more general applicability of the proposed approach?
* Regarding the statement in Section 5, the authors say that 'We establish that range regularization doesn’t affect the performance of the floating-point models as well, and in some overparameterized models, it also has a regularization effect to address overfitting.'. However, i think there is a lack of theoretical or experimental evidence supporting the claim that range regularization is more effective in preventing overfitting than previously proposed regularization techniques. Is there any theoretical basis or experimental contents related to this claim?

---

> ### Author Response · Authors · 2023-11-23
>
> Thank you for the helpful comments.
>
> While regularization (like L2-regularization) is a popular technique to improve model generalization during training, its final generalization capability (indicated by the val/test accuracy) after model compression may not be sufficient. In $R^2$, we are demonstrating that carefully-crafted and targeted regularization techniques (like our smin-max) can be helpful for the generalization AFTER compression (again indicated by the val/test accuracy) as shown in Table 2 in Section 4.
>
> To resolve your concern that we conducted comparison studies with $R^2$ using small models like MobileNet, Resnet18, and MobileBERT, we performed further comparison studies using larger models as you suggested. In the following tables, we can see that $R^2$ still shows significant accuracy improvement than without $R^2$ cases. From the large model quantization result in the tables and MobileBERT result in our paper, we believe that the $R^2$ would be effective for a large language model (LLM) as well. We are planning to apply $R^2$ in LLM quantization/compression with state-of-the-art techniques.
>
>
> * **AdaRound 4bit weight / 8bit activation all-layer quantization**
> |   Model  | W/O $R^2$ | R_inf | R_Margin | R_SMM |
> |----------|:---------:|:-----:|:--------:|:------|
> | Resnet50 |   70.51   | 71.00 |   74.38  | 72.55 |
> | Resnet101|   67.30   | 75.45 |   75.03  | 73.37 |
>
> * **SQuant 4bit weight / 8bit activation all-layer quantization**
> |   Model  | W/O $R^2$ | R_inf | R_Margin | R_SMM |
> |----------|:---------:|:-----:|:--------:|:------|
> | Resnet50 |    70.87  | 72.65 |   71.57  | 72.83 |
> | Resnet101|    68.34  | 73.11 |   74.98  | 74.63 |
>
>
> As you pointed, activation is not in our scope. PACT addressed outliers in activation by applying learnable clipping thresholds on activation during quantization. The reason why we aggressively quantize activation as well in our study is that we want to show $R^2$ does not ruin activation quantization even it changes the shape of weight distribution. If $R^2$ negatively affected activation quantization, then we will not be able to get better accuracy for weight and activation quantization. In our study, $R^2$ helps quantization accuracy even at low-bit for weight and activation together, so we think $R^2$ does not hurt activation quantization.

---

> ### Comment · Reviewer_aeAj · 2023-11-23
>
> Thank you for the detailed answers and results. I have read the authors' rebuttal as well as other reviews. However, I still find the supporting evidence or data for some of the claims made in the paper to be insufficient. So I would like to keep my rating.

---

### Official Review · Reviewer_vWZn · 2023-11-01

**Soundness:** 2 fair
**Presentation:** 3 good
**Contribution:** 2 fair
**Rating:** 5
**Confidence:** 4

**Summary:**

This paper proposes range regularization to shape the weight distribution for model compression and quantization. Three variants of range regularization are introduced. Experiments with ResNet-18, MobileNet-V1/2 (pretrained on ImageNet) are provided to demonstrate the effectiveness of range regularization on model compression and quantization.

**Strengths:**

The idea of shaping weight distribution for model compression and quantization makes sense. It seems the introduced R_inf and R_margin are effective to eliminate outliers from weight distributions.

The paper is well-written, and easy to follow.

**Weaknesses:**

It makes sense to eliminate outliers to improve the performance of weight quantization. However, it seems there is no explanation or justification why elimination of outliers can improve the performance of model compression.

The performance improvement of the proposed method is more pronounced for very aggressive compression ratio or 2-bit quantization, in which cases the baseline accuracies drop significantly (e.g., mostly halved). This makes the proposed method not very practical as it’s not acceptable in almost all cases to deploy a significantly worse model even though the compression rate is high. Often time, the compressed models should have similar or slightly worse performance to the original uncompressed models.

All model compression and quantization baseline methods are from 2019, 2020 or 2021. I didn’t follow the research in the area closely. But I believe there should be SOTAs in the past 2 years. So, it’s unclear how effective the proposed method is over the latest baselines.

Typos:
page 3 bottom: tate-of-the-art
page 6 middle: prove to be effective here.  -> ineffective

**Questions:**

As indicated above, a justification why weight shaping can help model compression would be helpful.

Please consider using the SOTAs from 2022 or 2023 as baselines. It would be interesting to see if the gains still hold for aggressive compression or quantization.

---

> ### Author Response · Authors · 2023-11-23
>
> We appreciate your valuable questions for our work.
>
> The compression algorithm used here (DKM) relies on k-means clustering. The centroids obtained in K-Means can be affected by outliers [Clustering88], since mean is affected by outliers.  Therefore, the centroids used in the compression algorithm are mis-aligned to the true weight representation possible with the given bit-width. We have claimed that such scenarios can be mitigated using $R^2$.
>
> We agree that the aggressive model quantization/compression shows noticeable accuracy regression comparing full precision model’s accuracy. However, we believe that it would be necessary to extremely quantize/compress a model in many cases such as on-device inference where we only have very limited computing resources. While many quantization works showed almost the same accuracy of a full-precision model at higher bit like 4bit or 8bit, we would like to focus on an extreme case, lower-bit and all-layer quantization/compression. With $R^2$, we can meaningfully reduce the accuracy gap between a quantized model and its full precision model at lower bit quantization such as 65.72% vs. 69.76% (Resnet18 2bit QAT), 57.36 vs. 73.08% (MobileNet-V2 2bit QAT), 72.29% vs. 74.12% (MobileNet-V1 W/A 4bit/8bit PTQ), and 66.10% vs. 69.76% (Resnet18 1bit compression). There would be trade-off between model accuracy and computing resource constraint. We will discuss more on this concern.
>
> As you suggested, we conducted a comparison study more especially for recent prior works such as PD-Quant [CVPR 2023] and SQuant [ICLR 2022]. The following table shows 4bit weight and 8bit activation quantization using the two quantization methods for MobileNet-V2. In the table, all $R^2$s improved quantization accuracy and Margin $R^2$ achieves the best performance. As we discussed in the paper, soft min max $R^2$ might be less effective for symmetric quantization as it makes weight distribution asymmetry.
> * **MobileNet-V2 4bit weight / 8bit activation all-layer quantization**
> | Quantization methods | W/O $R^2$ | R_inf | R_Margin | R_SMM |
> |----------------------|:--------:|:-----:|:--------:|:-----:|
> | PD-Quant [CVPR 2023] |   58.54  | 62.19 |   63.67  | 59.54 |
> | SQuant   [ICLR 2022] |   59.10  | 63.98 |   67.03  | 66.86 |
>
> Thanks for reviewing our paper in a very detailed manner. We will do proofreading again and fix typos in our manuscript.
>
>
> [Clustering88] Milligan, G.W., Cooper, M.C. A study of standardization of variables in cluster analysis. Journal of Classification 5, 181–204 (1988). https://doi.org/10.1007/BF01897163
>
> [PD-Quant] Liu, Jiawei, et al. "Pd-quant: Post-training quantization based on prediction difference metric." Proceedings of the IEEE/CVF Conference on Computer Vision and Pattern Recognition. 2023.
>
> [SQuant] Guo, Cong, et al. "SQuant: On-the-fly data-free quantization via diagonal hessian approximation." In International Conference on Learning Representations, (2022).

---

### Meta-Review · Area_Chair_V4UG · 2023-12-03

**Metareview:**

This paper is about model compression and quantization. While appreciating the novelty of the proposed technique, the reviewers raised various concerns, which aren’t fully addressed by the author’s rebuttal. Based on my own reading, I agree with the reviewers that the experimental results fail to support the main claims. Overall, I think the paper contains certain value but the current version cannot meet the standard of this conference. I would recommend rejecting the paper.

**Justification For Why Not Higher Score:**

While appreciating the novelty of the proposed technique, the reviewers raised various concerns, which aren’t fully addressed by the author’s rebuttal. Based on my own reading, I agree with the reviewers that the experimental results fail to support the main claims.

**Justification For Why Not Lower Score:**

N/A

---

### Decision · Program_Chairs · 2024-01-16

Reject